# Inertial Sensor Reliability and Validity for Static and Dynamic Balance in Healthy Adults: A Systematic Review

**DOI:** 10.3390/s21155167

**Published:** 2021-07-30

**Authors:** Nicky Baker, Claire Gough, Susan J. Gordon

**Affiliations:** Flinders Digital Health Research Centre, Flinders University, Adelaide, SA 5042, Australia; claire.gough@flinders.edu.au (C.G.); sue.gordon@flinders.edu.au (S.J.G.)

**Keywords:** inertial measurement unit, postural balance

## Abstract

Compared to laboratory equipment inertial sensors are inexpensive and portable, permitting the measurement of postural sway and balance to be conducted in any setting. This systematic review investigated the inter-sensor and test-retest reliability, and concurrent and discriminant validity to measure static and dynamic balance in healthy adults. Medline, PubMed, Embase, Scopus, CINAHL, and Web of Science were searched to January 2021. Nineteen studies met the inclusion criteria. Meta-analysis was possible for reliability studies only and it was found that inertial sensors are reliable to measure static standing eyes open. A synthesis of the included studies shows moderate to good reliability for dynamic balance. Concurrent validity is moderate for both static and dynamic balance. Sensors discriminate old from young adults by amplitude of mediolateral sway, gait velocity, step length, and turn speed. Fallers are discriminated from non-fallers by sensor measures during walking, stepping, and sit to stand. The accuracy of discrimination is unable to be determined conclusively. Using inertial sensors to measure postural sway in healthy adults provides real-time data collected in the natural environment and enables discrimination between fallers and non-fallers. The ability of inertial sensors to identify differences in postural sway components related to altered performance in clinical tests can inform targeted interventions for the prevention of falls and near falls.

## 1. Introduction

Postural control of balance is essential for keeping upright, moving effectively, and reacting to environmental challenges [1]. Good balance improves quality of life and wellbeing. Conversely, balance deficits can lead to a near fall or fall that may result in physical, psychological, or social consequences and, in some cases, death [2]. Near falls occur due to a loss of balance from a slip, trip or stumble where a fall is avoided “because a corrective action is taken to recover balance” [3] (p. 49). Although near falls are a predictor for falls [4], there is limited research concerning near falls, resulting in an unknown trajectory of the decline from near falls to falls [5]. People living in the community who have near falls and do not sustain an injury escape the attention of the health system. However, they are the group most likely to benefit from interventions to prevent falls. Until recently, having a fall has been the best predictor of having another fall. Recent evidence has identified clinical tests, namely single leg stance, lunge, and tandem walk five steps, that are able to discriminate near-fallers from fallers and non-fallers [6]. While gross changes in the performance of these tests are associated with falls history, there is no understanding of the contribution of postural sway to these outcomes.

Postural sway, the movement of the body over the base of support, is an indicator of balance. The traditional methods of measuring the speed, direction, and amplitude of postural sway by force plates or motion capture in gait laboratories has been superseded by wearable inertial sensors with recent interest in their measurement of standing balance [7] and gait [8]. Compared to the laboratory equipment, inertial sensors are inexpensive, portable, and permit measurements of postural sway to be taken in any setting specific to the population under investigation [9]. Additionally, wearable inertial sensors are small, lightweight, unobtrusive, and can be fixed on the body by tape, belt, or strap. Sensor data can be captured on three axes and can therefore provide detailed information in three dimensions of subtle changes in postural sway for static or dynamic conditions.

Inertial sensor measures of sway can discriminate between various age groups, and between healthy adults and adults with Parkinson’s disease [10], multiple sclerosis [11], and other neurological conditions [12]. Falls risk assessment by wearable inertial sensor is more sensitive than clinical testing using the timed up and go [13]. However, the reliability and validity of inertial sensors to measure postural sway is still unclear [14], especially in seemingly healthy populations without known pathology who experience near falls and falls.

Therefore, the aim of this systematic review was to examine and synthesize the current literature on the validity and reliability of wearable inertial sensors to measure postural sway in healthy adults undertaking static and dynamic balance tests.

## 2. Materials and Methods

### 2.1. Search Strategy

Three stages of searches were undertaken, following PRISMA guidelines [15]. The first stage was to identify systematic reviews that investigated ‘postural balance’, ‘inertial sensors’, and ‘reproducibility of results’ via the reference list of a scoping review of systematic reviews previously conducted [16]. This search identified five systematic reviews [3,12,14,17,18]. One further relevant systematic review [7] was published after the scoping review went to press. The critical appraisal of these six recent systematic reviews [3,7,12,14,17,18] was undertaken by two independent reviewers, with a third person to mediate in the case of disagreement. None of these reviews directly answered the aims of this study. Therefore, a new search was conducted as stage two.

The existing systematic reviews assisted the development of search strategies, terms, and dates. Three main concepts informed keywords, MeSH, and search terms: ‘postural control’, ‘inertial sensors’, and ‘validity/reliability’ (see Appendix A for full list of search terms). Relevant truncations and expansions were applied for each database, which included Medline, PubMed, Embase, Scopus, CINAHL, and Web of Science. The dates for searching were from January 2019 to January 2021. Searches were conducted by a research librarian experienced in conducting systematic reviews.

Selection criteria followed PICO (population, intervention, comparison, outcome) principles as follows: (P) healthy adults including healthy adults as a control group; (I) wearable inertial sensor to measure static and dynamic balance; (C) force plates, motion capture or other digital or clinical measure; (O) reliability, validity, accuracy. Exclusions were for papers published with children or non-human subjects, balance or equilibrium other than postural, postural alignment, pressure sensors, and studies that investigated only static or dynamic balance, not both. Papers published before 2010 were excluded on the basis of technological advances in sensor manufacture in the past 10 years. Smartphone use was excluded because of the need to hold a device in the hand, thereby altering natural arm movement for balance maintenance or recovery [19]. Moreover, the range of balance tests interpretable by phone does not incorporate novel balance tests, such as tandem walk and lunge [6]. Only primary investigation studies were incorporated, including conference proceedings if peer reviewed. Language was limited to English.

The third search examined the reference lists of the included studies and the six systematic reviews for relevant studies that fitted the inclusion criteria.

### 2.2. Eligibility, Quality and Data Extraction

Two independent reviewers screened titles and abstracts against selection criteria prior to full text review. A third author was available for arbitration but was not required. All search information was managed using Covidence systematic review software. Critical appraisal of the internal and external validity of included studies was undertaken using JBI critical appraisal checklist for analytical cross-sectional studies [20].

The first two authors extracted data from the first five studies into Excel and cross-checked for accuracy. The first author then extracted the remainder of the data, which were checked for thoroughness by the third author.

### 2.3. Data Pooling

Data pooling was multistage. Studies were initially grouped broadly to validity or reliability, then refined within these two contexts. Validity was categorized as concurrent (compared to gold standard), discriminant (able to distinguish between groups), and convergent (related to the clinical measure). Reliability was categorized as internal consistency (inertial sensor accurately measures postural sway) or test-retest reliability (sensor data replicates the results of the same postural sway activity in the same person at two timepoints). Balance activities were dichotomized to static or dynamic tests, then further refined to sort into the same measurement outcomes, e.g., single leg stance for static balance; timed up and go for dynamic. Finally, the outcome measures for validity and reliability were grouped, e.g., Pearson’s rho for validity; intraclass correlations for reliability. Heterogeneity was examined using τ2, I^2^ and Cochran’s Q statistic using the interpretations: τ2 = 0 suggests no heterogeneity, I^2^ values < 25, 26–50%, and >75% suggest low, moderate, and high heterogeneity respectively, and a significant Q statistic indicated that the studies do not share similar effects [21].

### 2.4. Statistical Analysis

Interrater agreement between two reviewers was captured at three stages, namely title/abstract screen, full text inclusion, and reference list inclusion. Rater agreement was analysed using Cohen’s kappa with agreement values interpreted as ≥0.81 excellent, 0.61–0.8 good, 0.41–06 fair and ≤0.4 poor [22]. For meta-analysis, homogeneity with balance activity, sensor location, and measurement outcome were required [23]. Where heterogeneity prevented meta-analysis, synthesis of the data was conducted.

## 3. Results

The search strategy identified 5430 articles. Following duplicate removal, as well as screening of titles, abstracts, and full text, 19 articles met the inclusion criteria. One paper repeated a previous study with different analysis and was therefore excluded [24] (see PRISMA flow diagram, Figure 1).

### 3.1. Study Characteristics

The 19 studies assessed static and dynamic balance in 1145 people, of whom 686 (59.9%) were healthy (see Table 1, Study Characteristics). Exclusively healthy populations were investigated in three studies: two in young adults [25,26] and the third in older adults [27], while healthy populations formed the control or comparison group in the remaining studies. Fallers were identified as a subject group in four studies [28,29,30,31]. The majority of papers investigated postural sway in neurological conditions, including Parkinson’s disease (PD) [32,33,34,35], multiple sclerosis (MS) [36,37], Huntington’s disease [38], progressive supranuclear palsy [33], muscular dystrophy [39], cerebellar ataxia [40,41], and a single case study of person with a stroke [42]. Only one musculoskeletal condition was investigated: anterior cruciate ligament reconstruction rehabilitation [43]. In Table 1, both the healthy and pathological groups are described for completeness.

Static balance activities varied by foot position (feet apart, together, tandem, semi-tandem or single leg stance), eye condition (open or closed), surface texture (firm or soft), length of time (from 10 s [25,30,39,42] to 3 min [29]), and some static activities also included perturbation (nudge or pull) [30]. Static balance was measured by the Romberg [30], Tinetti [38], or limits of stability [34] clinical tests.

Dynamic balance was assessed with postural transitions (sit to stand, stand to sit, transfer chair to chair), stepping (first step, step up), walking for a set time or distance, running, turning around, and jumping forward and sideways (Table 1). The outcome measures to evaluate these activities were varied. For example, gait was measured by step length, velocity, regularity, height, length, continuity, or symmetry; stride length or velocity; cadence and/or stance time. No single clinical test was used consistently. Dynamic clinical measures also included walking tests (timed up and go [31,33,36,37,42], 10 m walk [41], six-minute walk test [34], 25-foot walk test [37], and jumping (dynamic postural stability index (DPSI)) [25].

When both static and dynamic balance were assessed in the one balance test, they were measured by the Berg balance scale [28,31,42] and MiniBEST [31,34]. The time or distance within standardized tests differed between studies, e.g., in the TUG, the standard 3 m walking distance was increased to 7 m distance to provide more consistent data for gait parameters [36,37], and included additional single or dual tasks [31]. Static balance data from the sensors were analysed by multiple methods, most commonly root mean square (RMS) of acceleration, but also maximum, minimum, or mean of acceleration, jerk, various measures of velocity and Euclidian norm minus one (ENMO), providing challenges in grouping for meta-analysis. Dynamic balance similarly had multiple different analyses of step and stride length, stance time, cadence, and velocity.

### 3.2. Quality Assessment

All included papers were observational studies. Quality assessment used JBI analytical cross-sectional study critical appraisal checklist (see Table 2 Quality Assessment). All papers described the exposure, outcomes, and appropriate data analysis methods in detail. All studies but one [39] used standard, objective criteria. However, four studies lacked explicit selection criteria [27,29,38,39] and several provided no detail of the setting for the study [25,26,27,29,32,34,37,39,40,42,43], which impacts replicability. Five of the papers provided no identification or management of confounding factors [25,26,29,34,38], which impacts the trustworthiness of the results in these papers. Interrater agreement between two reviewers for screening, full text, and reference list selections was analysed using Cohen’s kappa, with a result of k = 0.805 interpreted as a good result.

### 3.3. Sensors

Sensor type, number, position, fixation, sampling frequency, and calibration methods differed between studies as outlined in Table 3. Only one study used a dual axis accelerometer (antero-posterior and mediolateral) [32] while the remainder used triaxial sensors, providing accelerometry data for the additional vertical plane. Thirteen studies also used inertial sensors with inbuilt gyroscopes providing further rotational velocity information [26,28,29,33,34,35,36,37,41,42,43]. The most common inertial sensors were Opals and XSens, where accelerometry data measured concurrent input from multiple sensors placed on the trunk and extremities. Various options for sensor body position and fixation were identified between studies. The preferred position for a sole sensor was on the lumbar spine [25,28,40,42] as this position corresponded closest to the centre of gravity of the body. Further, sensors situated on the low lumbar spine produced greater accuracy than thoracic sensors [27]. Studies using multiple sensor systems located them on the lower back, sternum, wrists, and ankles. Methods of fixation were not described in nine studies (47%). When stated, fixation from elasticated belts or bands [25,30,34,35,36,39,40] or adhesive tape [27,28,42] were the preferred methods. Two papers discussed movement artefacts [25,42]. However, only one excluded data due to sensor movement [42]. Sampling frequency ranged from 20 to 400 Hz, although Velazquez-Perez [41] provided no information on this. Only one paper [31] described down-sampling, which is the process of reducing the sample rate of a signal to manage the size of data. Accelerometry and gyroscopic data were analysed using sensor-specific tools [27,30], or in programs such as MATLAB [25,26,27,28,29,31,32,34,35,37,38,42,43] or Mobility Lab [33,34,36]. The statistical program ‘R’ was used in one study [39] and STASTICA in another [41].

### 3.4. Validity

The validity of the inertial sensor to measure balance was explored through concurrent (compared to gold standard), discriminant (able to distinguish between groups), and convergent (related to the clinical measure) validity. Data pooling was not possible for meta-analysis concerning validity due to the variety of protocols and outcome measures undertaken.

Concurrent validity was assessed by comparing the inertial sensor with force plates for static balance in six studies [25,26,28,29,32,35] and with force plates or motion capture systems for dynamic balance in three studies [35,38,42]. The resultant correlations identified that inertial sensors provide moderate to strong evidence of concurrent validity for medio-lateral (ML) (r = 0.58–0.84) [25,28,30] and antero-posterior (AP) sway (r = 0.71) [26] in static balance. There were good to excellent correlations between inertial sensor and instrumented walkway for step time (ICC 0.68–0.92), step length (ICC 0.68–0.89), and gait velocity (ICC 0.90–0.94) [25,38,42].

Discriminant validity was used to compare inertial sensor measures between young and older healthy participants [29,32,42], fallers and non-fallers [28,29,30,31], and healthy controls from people with specific diagnosed conditions [32,33,34,35,36,37,38,39,40,43]. Young adults showed significantly less medio-lateral sway than older adults during static stance [32,42]. The same was true for dynamic activities including gait velocity, step length, turning speed and stand to sit [42]. Inertial sensors were able to distinguish sway differences between fallers and non-fallers [28,29,30,31]. Dynamic balance activities to discriminate fallers from non-fallers included AP acceleration of walking [29,30], and functional activities of stepping on a stool and sitting to standing [30,31]. Inertial sensor classification accuracy for discriminating fallers from non-fallers ranged between 72.24% (95%CI 69.84–74.52%) [28] and 89% [30]. Compared to diagnostic populations, inertial sensors discriminated healthy controls by significantly reduced sway amplitude in eyes closed condition [32,33,34,37,40,42] and increased anticipatory postural adjustments [32,34,35,38,39] during standing. In dynamic balance, both walking cadence and turning velocity discriminated healthy controls [33,34,37]. Neither sit to stand nor stand to sit activities discriminated diagnostic groups from healthy controls. Discrimination accuracy varied between studies. There were moderate to strong results to discriminate healthy controls by sway acceleration amplitude in standing (AUC 0.68) [37], with eyes open or closed (classification accuracy 94–96%) [43] and lateral trunk range of motion in gait (AUC 0.72) [37].

Regarding sensor position, a single lumbar spine sensor identified significant differences between younger and older healthy adults [32,42] and was able to distinguish. the different dynamic balance tasks of lateral and forward jumps [25]. Further, the single sensor was as accurate as the six-sensor array [33].

Convergent validity evaluated the inertial sensor balance measures against clinical balance tools. The six-sensor array identified differences between the study group and healthy controls when observed, whereas timed clinical tests could not [33,37].

### 3.5. Reliability

Reliability was investigated in eight papers [25,26,27,29,34,35,36,39]. Internal consistency was assessed in three studies by evaluating test results across multiple sensors during the same activity (ICC 0.62 to 0.98) [26,27,39]. Inter-accelerometer reliability was good between right and left limbs (ICC > 0.8) [39], between L4 and L5 (r = 0.78–0.95) [27], between thoracic and lumbar spine (r = 0.60–0.76) [27] and between lumbar spine and ankle (inter-item correlation 0.70–0.98) [26], but not between upper and lower limbs (ICC = 0.59) [39].

Test-retest reliability showed reasonable consistency between studies. Meta-analysis was possible when static stance incorporated feet apart eyes open, measured by RMS of acceleration ML and AP, and when intraclass correlations were undertaken for statistical analysis [25,34,36]. Results from the grouped studies produced high homogeneity (I^2^ = 0.0%) with similar effects (Cochrane’s Q non-significant 0.14) indicating trustworthiness of the sensors to measure static balance (Figure 2). However, the lower quality of two of the included papers [25,34] (Table 2) influenced the strength of findings. Therefore, meta-analysis results were considered informative rather than conclusive. Measurements of static sway distance, sway area, path length, mean velocity, and RMS were reliable, indicated by moderate to good correlations ranging between ICC 0.57 and 0.79 [34,36]. Although dynamic balance was measured in diverse balance tasks, all test-retest parameters of dynamic balance produced moderate to excellent correlations (ICC 0.696–0.94), indicating strong correlations and good reliability in healthy adults [25,34,36].

## 4. Discussion

The aim of this systematic review was to investigate and synthesize the validity and reliability of wearable inertial sensors to measure postural sway in static and dynamic balance for healthy adults. Test-retest reliability results were consistently moderate to excellent for static and dynamic balance across the included studies. Meta-analysis was impossible for the validity studies due to heterogenous samples and methods. However, the synthesis showed moderate to good validity overall. These findings indicate consistency against gold standard equipment for measures of ML and AP sway in static balance and step time, step length, and gait velocity for dynamic balance. While the sensors were able to discriminate young from old, and fallers from non-fallers, the accuracy of discriminating healthy controls from diagnostic groups varied between studies.

The variability in equipment included multiple types of sensor. While all studies used accelerometer data, fewer included gyroscope data, suggesting data from accelerometers may be sufficient for clinical interventions. This reduced complexity may encourage more clinicians who are unfamiliar with the technical aspects of the new equipment to integrate sensors into practice. The multiple strategies for data acquisition, feature extraction, signal processing, and data analysis presented a heterogenous mix unsuitable for meta-analysis.

There was no consistent number of sensors or sensor placement position. However, the lumbar spine (L3–L5) was the preferred site overall. A single inertial sensor was as reliable as multiple sensors when placed near the centre of mass (L3–L5) and showed moderate to good validity and test-retest reliability for both static and dynamic balance. A single sensor placed over the centre of mass would provide simplicity in the clinical setting, particularly during telehealth interactions when instruction, observations, and interventions are provided remotely. A single sensor also aligns with recent literature for identifying differences between fallers and non-fallers [44]. However, using different placements for static and dynamic balance activities [29], and different body positions for sensor fixation, created challenges with pooling data. While different research questions demand different types of analysis, the standardization of the sensor position would permit a comparison of results across studies.

The validity of sway measures from wearable inertial sensors compared to the gold standard force plates or motion capture provided promising results across studies. These results concur with a previous scoping review of systematic reviews [16] as well as recent studies investigating the concurrent validity of sensors to measure balance in healthy adults [8,45,46,47]. In healthy populations, this indicates that inertial sensors provide valid data when used in home and community settings [48]. This provides flexibility for clinical treatment and trials, particularly in rural and remote settings, or during social distancing such as with COVID-19 [49]. Importantly it ensures that performance during testing is not altered by an unfamiliar environment. Therefore, these findings provide reassurance that the sensors are a valid proxy for the gold standard as a means of measuring static and dynamic balance in the community.

Sensors were valid in discriminating sway between younger and older participants, reinforcing the sway changes that occur due to ageing [50]. Sensors also discriminated fallers from non-fallers. The sensor data discriminated sub-tasks within clinical tests such as separating components for the timed up and go into the sit-to-stand, walk straight, turn, and stand-to-sit, which is consistent with previous findings in timed up and go [51]. While several studies measured the sway differences between fallers and non-fallers, no studies investigated differences between non-faller, fallers, and those who had experienced near falls. As this review provides evidence that sensors can identify subtle changes in sway between different aged healthy people, it is possible that sensors may identify sway differences between near fallers, fallers, and non-fallers. The early detection of subtle changes in postural sway is required to identify the risk of near falls [4,52] and can be measured reliably and with confidence of validity using inertial sensors.

The main limitation to this investigation was the inability to pool included studies for meta-analysis due to heterogeneity with balance activity, sensor location, and measurement outcomes. Additionally, some limitation may be considered from the inclusion of articles written only in English.

## 5. Conclusions

Measuring postural sway using inertial sensors in healthy adults permits assessment and treatment in the person’s natural environment, providing reassurance of accurate measures during times of social distancing. The ability to identify separate components of clinical tests using sensors permits the detection of subtle sway changes that may contribute to understanding sway differences for near falls as well as falls. Further research is required to evaluate the convergent validity of using a single sensor over the centre of mass rather than a six-sensor array for clinical balance tests such as the timed up and go test. Similarly, further research using a single sensor to discriminate sway differences between healthy and diagnostic groups, distinct age groups, and fallers/non-fallers would encourage the clinical uptake of sensors.

## Figures and Tables

**Figure 1 sensors-21-05167-f001:**
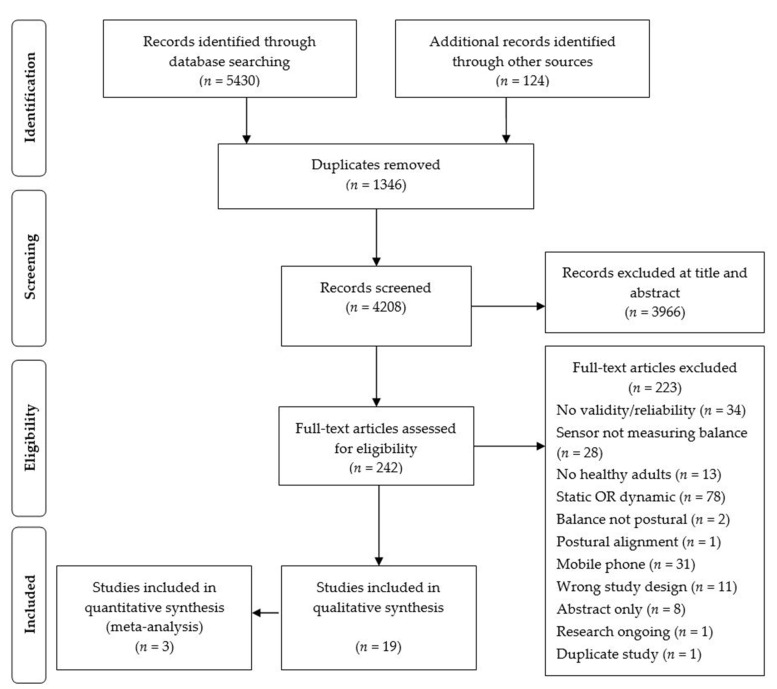
PRISMA flow diagram.

**Figure 2 sensors-21-05167-f002:**
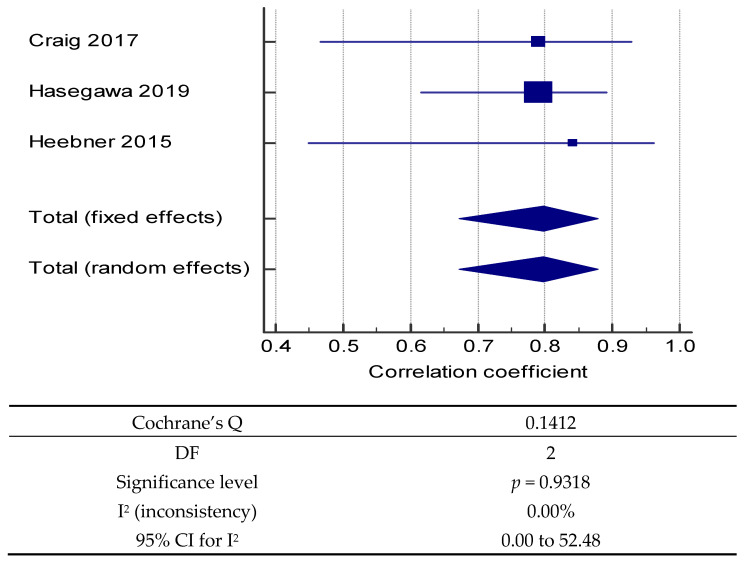
Meta-analysis.

**Table 1 sensors-21-05167-t001:** Study Characteristics.

Author, Year, Setting, Country [Reference]	Study Population, Number (Sex) Age in Years Mean ± SD (Range)	Healthy Group Number (Sex) Age in Years Mean ± SD (Range)	Static Balance Activity	Dynamic Balance Activity	Clinical Balance Measure	Outcome Measure
Bzduskova et al., 2018, NA, Slovak Republic, [32]	PD *n* = 13 (8M 5F) 63.7 ± 5.7 y	Young *n* = 13 (4M 9F) 25.0 ± 2.3 y Older *n* = 13 (4M 9F) 70.1 ± 4.5 y	FA EO, FA EC	Step with vibration	NA	RMS acc AP ML, jerk, mean veloc, peak veloc, stride length, stride veloc, cadence, stance time
Craig et al., 2017, Lab, USA [36]	MS *n* = 15 (3M 12F) 48.2 ± 8.7 y	HC *n* = 15 (3M 12F) 47.8 ± 9.5 y	FA EO	7 m TUG	TUG	RMS acc ML AP V
Dalton et al., 2013, Lab, Wales, UK [38]	HD Pre-manifest *n* = 10 (4M 6F) 44.8 ± 11.7 y Manifest *n* = 14 (8M 6F) 51.8 ± 14.8 y	HC *n* = 10 (5M 5F) 56.4 ± 10.9 y	FT EO, FT EC	5 m walk	Romberg	ENMO
De Vos et al., 2020, Lab, England, UK [33]	PSP *n* = 21 (12M 9F) 71 y (63–89) PD *n* = 20 (11M 9F) 66.4 y (50–79)	HC *n* = 39 (19M 20F) 67.1 y (51–82)	FA EC	TUG, 2 min walk	TUG	min, max, mean acc AP ML V
Greene et al., 2012, Hospital clinic, Ireland [28]	Fallers *n* = 100 (NA) Whole study (57M 63F) 73.3 ± 5.8 y	Non-faller *n* = 20 (NA)	Semi TS EO 40 s, FT EC 30 s, Turn head	STS, stand to sit, transfer, fwd reach, pick up object, turn 360, place foot on stool	BBS	Peak accel, jerk, stride length, stride veloc, cadence, stance time.
Hasegawa et al., 2019, NA, USA [34]	PD *n* = 144 (93M 51F) 68.4 ± 8.0 y	HC *n* = 79 (48M 31F) 68.2 ± 8.1 y	FT EO, FT EC, FT EO soft, FT EC soft, LOS, APA, APR	Step, ISAW, ISAW single task, ISAW dual task	ISAW MiniBEST	RMS acc ML AP; cadence.
Heebner et al., 2015, NA, USA [25]	NA	Healthy Reliability *n* = 10 (10M 0F) 24.3 ± 4.2 y Validity *n* = 13 (13M 0F) 24.1 ± 3.1 y	FA EO, FA EC, FAEO soft, FAEC soft, TS EO, TS EC, SLS EO, SLS EC.	DPSI-AP, DPSI-ML	NA	RMS acc AP ML, mean acc AP ML, stride length, stride veloc, stance time.
Jimenez-Moreno et al., 2019, NA, England UK [39]	MD *n* = 30 (20M 10F) 48 y (25–72)	HC *n* = 14 (6M 8F) 32 y (23–47)	FA EO	6 minWT, 10 mWT, 10 m Walk/Run Test	6 minWT	Peak trunk veloc sagittal.
Leiros-Rodriguez et al., 2016, NA, Spain [27]	NA	*n* = 66 (0M 66F) 64.9 ± 7.6 y	SLS EC, SLS EO soft	walk 10 m, turn, walk 10 m	NA	RMS acc ML AP, stride length, cadence.
Liu et al., 2012, NA, USA [29]	Fallers *n* = 4 (2M 2F) 74.5 ± 2.7 y	Young *n* = 4 (1M 3F) 21.8 y ± 1.0 y Older *n* = 4 (2M 2F) 73.3 ± 7.1 y	FA EO, FT EO, FA EC 10 s	Treadmill walk	NA	RMS acc AP ML V, jerk, sway area, path length, mean velocity, cadence.
Mancini et al., 2016, Lab (validity), clinic (reliability) USA [35]	PD Validity *n* = 10 (8M 2F) 67.2 ± 5 y Reliability *n* = 17 (12M 5F) 67.1 ± 7.0 y	HC Validity *n* = 12 (9M 3F) 68.0 ± 5.0 y Reliability *n* = 17 (6M 11F) 67.9 ± 6.0 y	FA EO	APA, first step, walk	NA	Peak acc ML AP, angular veloc, APA duration, step length, step velocity.
Martinez-Mendez et al., 2011, NA, Japan [26]	NA	*n* = 10 (7M 3F) 26 ± 3 y	FA 2 cm EO	APA, step fwd	NA	RMS acc AP ML, peak acc AP ML, sway area, jerk, trunk veloc sagittal, stride length, stride veloc, stance time, cadence.
Matsushima et al., 2015, NA, Japan [40]	SCA or CA *n* = 51 (24M 27F) 60.3 ± 10.4 y	HC *n* = 56 (28M 28F) 57.2 ± 14.1 y	FA EO, FA EC, FT EO, FT EC	walk 10 m	NA	VM horizontal acc; gait velocity, cadence, step length, step regularity, RMS ratio.
O’Brien et al., 2019, NA, USA [42]	Stroke *n* = 1 (1M 0F) 57 y	Young *n* = 14 (8M 6F) 26.4 ± 3.9 y Middle *n* = 19 (8M 11F) 43.7 ± 5.8 y Older *n* = 16 (8M 8F) 61.8 ± 5.1 y	FA EO, FA EC, FT EO, TS EO, SLS EO	10 mWT normal veloc, 10 mWT high veloc, TUG	BBS TUG	Max/mean acc AP ML V, stride length.
Rivolta et al., 2019, Rehab Centre, Italy [30]	Inpatient Fallers *n* = 33 (26M 7F) 72.7 ± 15.2 y	Inpatient *n* = 46 (30M 16F) 72.5 ± 11.5 y Volunteers *n* = 11 (0M 11F) 35.7 ± 14.0 y	FA EO, FA EC, FA EC nudge	360° turn, walk 10 m, sit to stand, stand to sit	Tinetti test	RMS acc AP ML V; mean acc AP ML V; VM; step height/length/symmetry/continuity, trunk sway.
Senanayake et al., 2013, NA, Brunei Darussalam [43]	ACLR rehab *n* = 8 (6M 2F) 31.0 ± 4.1 y	HC *n* = 4 (3M 1F) 31.0 ± 8.3 y	SLS EO, SLS EC	Treadmill 4 kph; Treadmill 6 kph	NA	RMS acc AP ML.
Spain, St George et al., 2012, NA, USA [37]	MS *n* = 31 (12M 19F) 39.8 y (24–67)	HC *n* = 28 (9M 19F) 37.4 y (26–60)	FA EO, FA EC	T25FW, 7 m TUG	ABC, MSWS12, EDSS TUG	RMS accel AP ML, jerk, mean/peak/sway veloc, stride length, cadence, turning time, trunk rotation.
Tang et al., 2019, Uni, USA [31]	Fallers *n* = 14 Whole study *n* = 30 (13M 17F) 76.0 ± 10.5 y	Non faller *n* = 16 (NA)	FA EO	MiniBEST including TUG and dual task TUG; BBS	BBS, MiniBEST, TUG	Peak acc AP ML V, cadence, stride/step/swing, stance time.
Velazquez-Perez et al., 2020, research centre, Cuba [41]	SCA *n* = 30 (7M 23F) 43.5 ±10.5 y	HC *n* = 30 (7M 23F) 43.3 ± 10.2 y	FA EO FT, TS	10 m walk, Tandem walk 10 steps		

Key: ABC Activities-specific Balance Confidence; acc acceleration; ACL anterior cruciate ligament reconstruction; AP antero-posterior; APA anticipatory postural adjustment; APR automatic postural response; BBS Berg Balance Scale; BEST Balance Evaluation Systems Test; CA cerebellar ataxia; DPSI dynamic postural stability index (jump landing one leg); EC eyes closed; EDSS Expanded Disability Status Scale; ENMO Euclidean Norm Minus One; EO eyes open; FA feet apart; FT feet together; fwd forward; HC healthy controls; HD Huntington’s Disease; ISAW instrumented stand and walk test; Lab motion analysis laboratory; m meter; min minute; MD myotonic dystrophy; ML mediolateral; MS Multiple Sclerosis; MSWS12 MS Walking Scale (12 item); NA Not available; PD Parkinson’s Disease; PSP Progressive Supranuclear Palsy; Rehab rehabilitation; RMS root mean square; ROM range of motion; s second; SCA spinocerebellar ataxia; SLS single leg stance; STS sit to stand; TS tandem stance; TUG timed up and go test over 3 m; T25FW Timed 25 Foot Walk; Uni university; V vertical; veloc velocity; VM vector magnitude; WT walk test; y years of age.

**Table 2 sensors-21-05167-t002:** Quality Assessment–JBI cross-section study.

Author, Year, [Reference]	Inclusion Criteria Defined	Subject, Setting Described	Exposure Valid Reliable	Objective Standard Criteria	Confounders Identified	Confounder Strategies	Outcomes Valid Reliable	Appropriate Stats Analysis
Bzduskova, 2018 [32]	+	-	+	+	+	+	+	+
Craig, 2017 [36]	+	+	+	+	+	+	+	+
Dalton, 2013 [38]	-	+	+	+	-	-	+	+
De Vos, 2020 [33]	+	+	+	+	+	+	+	+
Greene, 2012 [28]	+	+	+	+	+	+	+	+
Hasegawa, 2019 [34]	+	-	+	+	-	-	+	+
Heebner, 2015 [25]	+	-	+	+	-	-	+	+
Jimenez-Moreno, 2019 [39]	-	-	+	-	+	+	+	+
Leiros-Rodriguez, 2016 [27]	-	-	+	+	+	+	+	+
Liu, 2012 [29]	-	-	+	+	-	-	+	+
Mancini ^1^, 2016 [35]	+	+	+	+	+	+	+	+
Martinez-Mendez, 2011 [26]	+	-	+	+	-	-	+	+
Matsushima, 2015 [40]	+	-	+	+	+	+	+	+
O’Brien, 2019 [42]	+	-	+	+	+	+	+	+
Rivolta, 2019 [30]	+	+	+	+	+	+	+	+
Senanayake, 2013 [43]	+	-	+	+	+	+	+	+
Spain, 2012 [37]	+	-	+	+	+	+	+	+
Tang, 2019 [31]	+	+	+	+	+	+	+	+
Velazquez-Perez, 2020 [41]	+	+	+	+	+	+	+	+

^1^ Both reliability and validity sub-studies.

**Table 3 sensors-21-05167-t003:** Sensors Overview.

Reference, Year	Sensor Type (Brand)	Number, (Body Location), Fixation	Sampling Frequency	Variables	Data Analysis Tool
Bzduskova et al., 2018 [32]	Dual axis accel (ADXL202)	2, (T4, L5), NS	100 Hz	Low pass filtered; cut-off frequency 5 Hz; Butterworth filter; calibration for ±30° range body tilt	MATLAB software
Craig et al., 2017 [36]	Triaxial accel/gyro (Opal)	6, (sternum, L5, bilat wrists, bilat ankles), elastic straps	128 Hz	Accel ranges ± 16 g, ±200 g; gyro range ±2000 deg/s	Mobility Lab software (APDM)
Dalton et al., 2013 [38]	Triaxial accel (AD-BRC)	1, (sternum), NS	250 Hz	Range ± 2.5–10 g, calibration by rotation through established angles; high pass filtered, 3rd order normalized elliptical filter, passband frequency 0.25 Hz	MATLAB software
De Vos et al., 2020 [33]	Triaxial accel/gyro (Opal)	6, (sternum, L5, bilat wrists, bilat feet), NS	100 Hz	Wireless data stream to laptop	Mobility Lab software
Greene et al., 2012 [28]	Triaxial accel/gyro (SHIMMER)	1, (L3), adhesive tape	102.4 Hz	Calibration using standard method; data streamed via Bluetooth to laptop	MATLAB
Hasegawa et al., 2019 [34]	Triaxial accel/gyro (Opal)	8, (sternum, L5, bilat wrists, bilat shins, bilat feet), elastic straps	128 Hz	Unscented Kalman Filter	Mobility Lab (APDM) and MATLAB
Heebner et al., 2015 [25]	Triaxial accel (ADXL78)	1, (L5), neoprene belt	100 Hz	Range ± 16 g, built in data acquisition and storage, low pass filter 50 Hz	MATLAB
Jimenez-Moreno et al., 2019 [39]	Triaxial accel (GENEActiv)	4, (bilat wrists, bilat ankles), elastic band	100 Hz	Output metric ENMO–mg.	R software
Leiros-Rodriguez et al., 2016 [27]	Triaxial accel (GT3 Plus)	3, (T4, L4, L5), adhesive tape	100 Hz	Configured 1 s timeframe. Concurrent analysis video & accelerometry data; reviewed analysis.	ActiLife software
Liu et al., 2012 [29]	Triaxial accel/gyro (MTX Xsens)	2, (L5, ankle), NS	50 Hz	Maximum Lyapunov exponent	MATLAB
Mancini et al., 2016 [35]	Triaxial accel/gyro (Opal validity; MTX Xsens reliability)	6 validity/3 reliability (sternum, L5, bilat wrists, bilat ankles) elastic straps	128 Hz Opal; 50 Hz MTX Xsens	3.5 Hz cut-off, zero-phase, low-pass Butterworth filter. Resampling from inertial sensor, force platform and infrared cameras at 50 Hz.	MATLAB
Martinez-Mendez et al., 2011 [26]	Unit with triaxial accel (MMA, Freescale) & gyros (X3500 Epson; ENC-03RC Matura)	2, (L3/4, ankle of dominant foot), NS	100 Hz	Accel range ± 1.5 g, gyro range ± 80 deg/s; response freq 0.01–58 Hz. Bluetooth transmission	MATLAB
Matsushima et al., 2015 [40]	Triaxial accel (Jukudai Mate)	1, (L3), elastic belt	20 Hz	Detection range ± 10 g; resolution power 0.02 g	BIMUTAS II
O’Brien et al., 2019 [42]	Triaxial accel/gyro (BioStampRC)	1, (L5), Tegaderm adhesive film	31.25 Hz	Accel ± 4 g; gyro ± 2000 deg/s; 4th order low pass Butterworth filter 2 Hz; acquisition with BioStampRC	MATLAB
Rivolta et al., 2019 [30]	Triaxial accel (GENEActiv)	1, (chest), elastic band	50 Hz	12 bits over range ± 8 g; chronometer for starting time; high pass 3rd order Butterworth filter	Manually segmented accel signals; GENEActiv software
Senanayake et al., 2013 [43]	Triaxial accel/gyro (KinetiSense)	4, (bilat thighs, bilat shins), NS	128 Hz	Wireless transmission via USB	KinetiSense and MATLAB
Spain, St George et al., 2012 [37]	Triaxial accel/gyro (XSens)	6, (sternum, L5, bilat wrists, bilat ankles), NS	50 Hz	Accel range ± 1.7 g; gyro range ± 300 deg/s. Filtered with 3.5 Hz cutoff, zero phase, low pass Butterworth filter	MATLAB
Tang et al., 2019 [31]	Triaxial accel (ADXL330)	2, (hip, foot), NS	400 Hz, down sampled to 25 Hz	Common and Activity Specific features extracted; mRMR feature selection	MATLAB
Velazquez-Perez et al., 2020 [41]	Triaxial accel/gyro (Opal)	6, (Hands, feet, sternum, L5), NS	NS	NS	STATISTICA

Key: accel accelerometer; deg degrees rotation; freq frequency; g gravitational velocity (m/s^2^); gyro gyroscope; Hz hertz; mRMR minimal-redundancy-maximal-relevance; NS not stated; s second.

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
