# Peer review of "Inertial Sensor Reliability and Validity for Static and Dynamic Balance in Healthy Adults: A Systematic Review"

_sensors, 2021, doi:10.3390/s21155167_

Round 1

Reviewer 1 Report

The systematic review was conducted following state of the art guidelines. The authors provided all the necessary information for understanding both procedures and results. The discussion was carried out thoroughly an the introduction is complete. The topic is interesting given the huge problem of risk pf falls in elderly and the need to provide alternative measures to gait lab measures. 

Minor comments 

Abstract

Please rephrase 

“Measuring postural sway using inertial sensors in healthy adults allows us reliably and validly to measure variations in postural sway and permits assessment in the persons natural environment”

“…changes that may contribute to falling and near falls.”

Author Response

Comment

Response

Abstract: please rephrase “measuring postural sway using inertial sensors in healthy adults allows us reliably and validity to measure variations in postural sway and permits assessment in the person’s natural environment”

“Using inertial sensors to measure postural sway in healthy adults provides real-time data collected in the natural environment and enables   discrimination between fallers and non-fallers.”

Please rephrase “…changes that may contribute to falling and near falls.”

 “The ability of inertial sensors to identify differences in postural sway component related to altered performance of clinical tests can inform targeted interventions for prevention of falls and near falls.”

Reviewer 2 Report

Please see the attached document

Author Response

Comment

Response

The authors mentioned that the previous existing systematic reviews, which did not answer the aims of this study, assisted in the development of the search strategies, terms and dates. However, the authors should specify why the articles published before 2010 were excluded.

Thank you, Added (line 92)

“Papers published before 2010 were excluded on the basis of technological advances in sensor manufacture in the past 10 years.”

The studies included in this review measured both static and dynamic balance. Nevertheless, static and dynamic balance are presented separately in this study.

Thank you. Results for static and dynamic balance are presented separately to describe and make explicit the differences in validity and reliability results. In the discussion the results are brought together.

The conclusion is quite concise. Based on the selected study, the authors can include some

suggestions for future studies.

The conclusion has been expanded, commencing line 368:

“Measuring postural sway using inertial sensors in healthy adults permits assessment and treatment in the person’s natural environment, providing reassurance of accurate measures during times of social distancing. The ability to identify separate components of clinical tests using sensors permits detection of subtle sway changes that may contribute to understanding sway differences for near falls as well as falls. Further research is required to evaluate the convergent validity of using a single sensor over the centre of mass rather than 6-sensor array for clinical balance tests such as the Timed Up and Go Test. Similarly, further research of a single sensor to discriminate sway differences between healthy and diagnostic groups, distinct age groups and fallers/non-fallers would encourage the uptake of clinical use of sensors.”

Page 7, line 160 (Results): please add “from” before “10 seconds”

Added (line 172)“time (from 10 seconds [25,30,39,42] to 3 minutes…”

Page 7, lines 158-171 (Results): In this part of the manuscript, I would keep the results from the static and dynamic activities separated in two paragraphs. Please consider moving the sentence “Static balance was measured…”
(lines 167-169) at the end of the previous paragraph.

Thank you, this has been moved to the previous paragraph (line 172)

“…some static activities also included perturbation (nudge or pull) [30]. Static balance was measured by the Romberg [30], Tinetti [38], or Limits of Stability [34] clinical tests.

Dynamic balance was assessed…”

Page 12, lines 229-232 (Results): the authors stated that  “inertial sensors provide moderate to strong evidence of concurrent validity” also for the antero-posterior sway. Nevertheless, the reported Pearson’s coefficient ranged from -0.536 to 0.71 (from negative to positive values).

This has been revised (line 244)

“moderate to strong evidence of concurrent validity for medio-lateral (ML) (r = 0.58-0.84) [25,28,30] and antero-posterior (AP) sway (r = 0.71) [26] in static balance”

Page 12, lines 239-241 (Results): Please revise this sentence.

Thank you, the results are re-written for clarity.

Page 13, lines 284-286 (Discussion): If I am not mistaken, a similar sensor placement (lower back) was adopted in the studies included in the meta-analysis. This is an information that could be added here.

Thank you, added line 334

“centre of mass (L3-5) and showed moderate to good validity and test-retest reliability for both static and dynamic balance.”

Reviewer 3 Report

Abstract:

Nineteen studies met the inclusion criteria

Meta-analysis: Is this a systematic review like indicated a few lines above or a Meta-alaysis. Be precise.

Keywords:

Only use Keywords that are not included in the title.

Introduction:

Line 28: Near falls occur due to a loss of balance

Line 47: Sensor data can be captured on three planes - sagittal, frontal, and transverse : Relating the sensor axis to sagittal, frontal and transverse is not correct and also misleading. The sensor coordinatesystem has three axis and data can be collected in three dimensions. But there is no relationship to body planes.

Line 58: 'Therefore, the aim of this systematic review was to investigate the validity and reliability of wearable inertial sensors to measure postural sway in healthy adults undertaking static and dynamic balance tests.' It it maybe just the way it is stated, but this article is not investigating the validity, it summarizes and analyses studies that have investigated validity etc.

Results:

Overall the results are hard to read because of the density of the data (which is not a bad thing). It would help if there would be a nicer way of displaying the resulst other than just plain text. 

Line 209: Two papers discussed movement artefacts [25,42]

I don't see the benefit of doing a meta-analysis on three papers. For many other results there were also three or four studies, and they were not pooled. In the end, the so-called meta-analysis is used as the data the conlcusion is built on, but these three studies are just used as they use the same methods, not because they are of especially high quality, or because they are representive for the work that is done in the field. 

I think it ould be a good idea to see whether the validity/reliabilty is related to the quality of the paper. 

Discussion:

It is questionable whether the conclusions drawn from the meta-analysis are reliable. As stated above, the criterion for being included in the meta-analysis is (almost) random, not reflecting the quality of the data/sample size etc. 

I would recommend removing the meta-analysis part of the study.

Line 313:  of a systematic review

References:

Line 53ff: delete the dummy references

Author Response

Comment

Response

Abstract: Nineteen studies met the inclusion criteria

Thank you, this has been altered (line 13)

“Nineteen studies met the inclusion criteria.”

Abstract. Meta-analysis: Is this a systematic review like indicated a few lines above or a Meta-alaysis. Be precise.

Thank you. This is a systematic review with meta-analysis being conducted when it was possible to pool the studies’ results. The sentence has been changed (line 14)

 “Meta-analysis was possible for reliability studies only and found that inertial sensors are reliable to measure static standing eyes open.”

Introduction. Line 28: Near falls occur due to a loss of balance

Thank you, this has been altered (line 35)

“Near falls occur due to a loss of balance…”

Introduction. Line 47: Sensor data can be captured on three planes - sagittal, frontal, and transverse : Relating the sensor axis to sagittal, frontal and transverse is not correct and also misleading. The sensor coordinate system has three axis and data can be collected in three dimensions. But there is no relationship to body planes.

Thank you. This has been revised (line 53)

“Sensor data can be captured on three axes, therefore can provide detailed information in three dimensions of subtle changes in postural sway for static or dynamic conditions.”

Introduction. Line 58: 'Therefore, the aim of this systematic review was to investigate the validity and reliability of wearable inertial sensors to measure postural sway in healthy adults undertaking static and dynamic balance tests.' It it maybe just the way it is stated, but this article is not investigating the validity, it summarizes and analyses studies that have investigated validity etc.

Revised, (line 64)

“Therefore, the aim of this systematic review was to examine and synthesize the current literature on validity and reliability of wearable inertial sensors to measure postural sway in healthy adults undertaking static and dynamic balance tests.”

Results. Overall the results are hard to read because of the density of the data (which is not a bad thing). It would help if there would be a nicer way of displaying the resulst other than just plain text. 

Results have been rewritten for clarity, commencing line 136.

Results. Line 209: Two papers discussed movement artefacts [25,42]

Line 224 “Two papers discussed movement artefacts.”

Results. I don't see the benefit of doing a meta-analysis on three papers. For many other results there were also three or four studies, and they were not pooled. In the end, the so-called meta-analysis is used as the data the conlcusion is built on, but these three studies are just used as they use the same methods, not because they are of especially high quality, or because they are representive for the work that is done in the field. 

Meta-analysis was only conducted when there was “homogeneity with balance activity, sensor location and measurement outcome” (line 131).

However, as you point out, the quality of two of the three papers able to be pooled for meta-analysis did not score well; therefore, cannot be fully relied upon. This has been reflected (line 295):

“However, the lower quality of two of the included papers [25, 34] (table 2) influenced the strength of findings, therefore meta-analysis results were considered informative rather than conclusive.”  

In all other cases there were dissimilar balance activities, sensor locations and outcome measures which precluded meta-analysis.

Results. I think it ould be a good idea to see whether the validity/reliabilty is related to the quality of the paper. 

Thank you, the impact of the papers’ quality is addressed in the results (line 295)

Discussion. It is questionable whether the conclusions drawn from the meta-analysis are reliable. As stated above, the criterion for being included in the meta-analysis is (almost) random, not reflecting the quality of the data/sample size etc. 

The discussion has been revised to provide further clarity, implications and relevance to the results, commencing line 313.

Discussion. I would recommend removing the meta-analysis part of the study.

Thank you. The meta-analysis has remained in the results and the quality of the papers impacting the influence of meta-analysis results is also described. Given only one of three reviewers has suggested removing the meta-analysis, we have decided to include it  

Discussion. Line 313:  of a systematic review

Line 346 “…concur with a previous scoping review of systematic reviews.”

References. Line 53ff – delete the dummy reference

Deleted
